# Global gridded population datasets systematically underrepresent rural population

Josias Láng-Ritter [1,2] ✉, Marko Keskinen [1] & Henrikki Tenkanen [2]

Numerous initiatives towards sustainable development rely on global gridded population data. Such data have been calibrated primarily for urban environments, but their accuracy in the rural domain remains largely unexplored. This study systematically validates global gridded population datasets in rural areas, based on reported human resettlement from 307 large dam construction projects in 35 countries. We find large discrepancies between the examined datasets, and, without exception, significant negative biases of −53%, −65%, −67%, −68%, and −84% for WorldPop, GWP, GRUMP, LandScan, and GHS-POP, respectively. This implies that rural population is, even in the most accurate dataset, underestimated by half compared to reported figures. To ensure equitable access to services and resources for rural communities, past and future applications of the datasets must undergo a critical discussion in light of the identified biases. Improvements in the datasets' accuracies in rural areas can be attained through strengthened population censuses, alternative population counts, and a more balanced calibration of population models.

The accurate estimation of population distribution is a central aspect of many scientific, social, and environmental endeavours, ranging from resource allocation[1,2] and infrastructure planning[3–5] to disease epidemiology[6] and disaster risk management[7–11]. In recent years, the advancement of geospatial technologies and the widespread availability of satellite imagery and remote sensing data have facilitated the development of global gridded population data[12,13]. These comprehensive datasets partition the planet into evenly spaced, high-resolution grid cells with population counts, enabling researchers and policy makers to gain insights into the spatial distribution of human populations on a global scale (Fig. 1).

To date, eight open-access datasets of population counts are available with (near-)global coverage, namely GWP (Gridded Population of the World)[14], GRUMP (Global Rural-Urban Mapping Project)[15], GHS-POP (Global Human Settlement Population)[16], LandScan[17], WorldPop[18], HYDE (History database of the Global Environment)[19], HRSL (High Resolution Settlement Layer)[20], and Kontur[21] (Table 1). The models behind these products have varying degrees of complexity, ranging from simple areal disaggregation of census counts (as in

GWP[14]) to dasymetric mapping approaches involving numerous auxiliary data sources, such as satellite-based detection of infrastructures and nightlights (as in WorldPop[18]). Details of the population datasets and their underlying methods are documented by Leyk et al.[12], TReNDS[13], and on the website of the POPGRID data collaborative (https://www.popgrid.org).

Due to their large spatial coverage and relevance for countless disciplines, the use and application of global gridded population datasets has dramatically increased in recent years[22], but a consistent global-scale assessment of their accuracy is to date lacking. The datasets have been primarily validated and assessed in scattered countries or regions[22–24], or focusing on selected urban areas, where population density is relatively high and the availability of ground-truth data is more accessible[25]. Conversely, rural areas, characterised by dispersed and heterogeneous populations, present unique challenges for population estimation due to limited ground-based data and inherent spatial complexities[12,13,26]. As a result, the accuracy and reliability of these datasets in rural regions remain largely unexplored, leading to a

¹Water and Development Research Group, Department of Built Environment, Aalto University, Espoo, Finland. ²GIScience for Sustainability Transitions Lab, Department of Built Environment, Aalto University, Espoo, Finland. ✉e-mail: josias.lang-ritter@aalto.fi

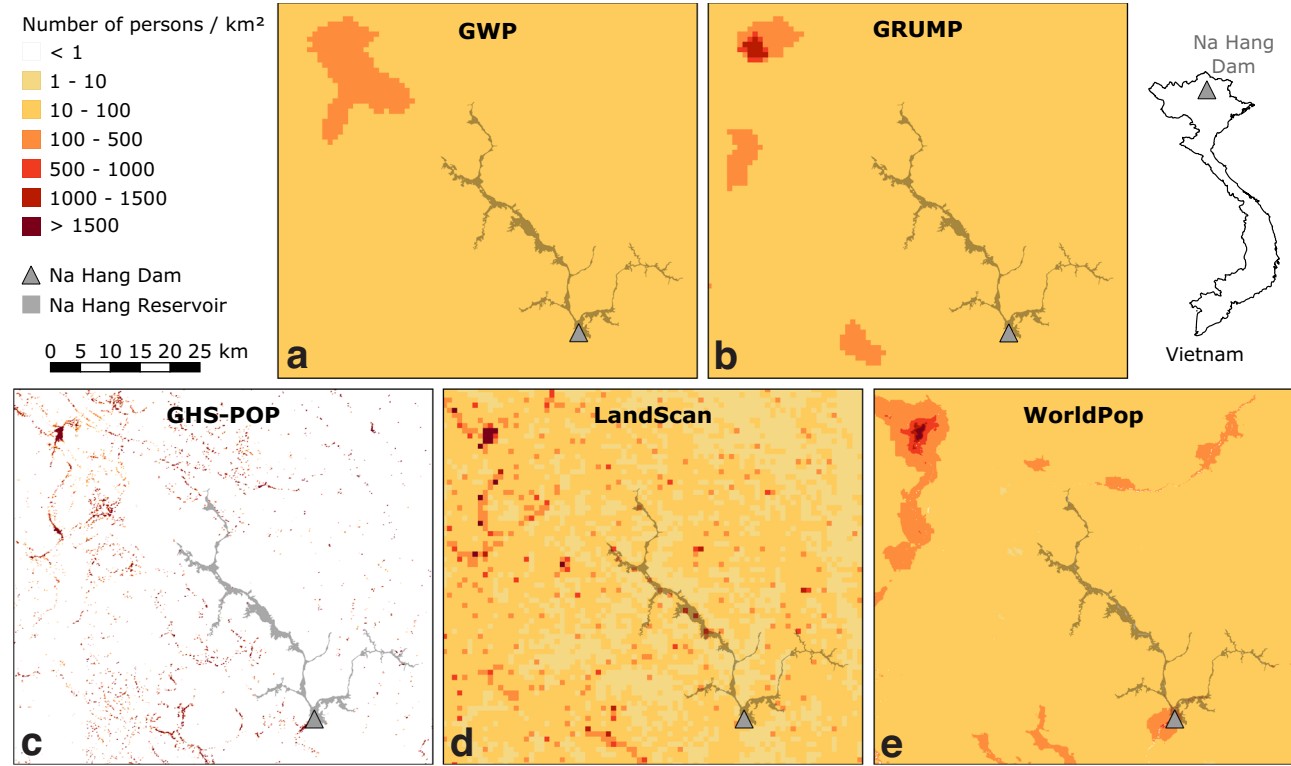

**Fig. 1 | Examples of five global gridded population datasets.** The map shows part of the rural province of Tuyên Quang in northern Vietnam, with population data for the reference year 2000 from **a** GWP, **b** GRUMP, **c** GHS-POP, **d** LandScan, and **e** WorldPop. The Na Hang Reservoir in this area (indicated by the grey polygon) was completed in 2008 and caused resettlement of 4000 people. Supplementary Fig. 1 shows an enlargement of panel d. Country boundary courtesy of ©EuroGeographics.

## Table 1 | Characteristics of global gridded population datasets

| Dataset name | Reference year(s) | Grid cell size | Data description | Data access (retrieved on 21.06.2023) | Included in this study |
|---|---|---|---|---|---|
| GWP | 2000–2020 | 30 arc-s (~1 km) | Doxsey-Whitfield et al.[14] | CIESIN (https://sedac.ciesin.columbia.edu/data/collection/gpw-v4) | Yes |
| GRUMP | 1990–2000 | 30 arc-s (~1 km) | Balk[15] | CIESIN (https://sedac.ciesin.columbia.edu/data/collection/grump-v1) | Yes |
| GHS-POP | 1975–2030 | 3 arc-s (~100 m) | Joint Research Centre[16] | Joint Research Centre (https://ghsl.jrc.ec.europa.eu/download.php?ds=pop) | Yes |
| LandScan | 2000–2020 | 30 arc-s (~1 km) | Dobson et al.[17] | Oak Ridge National Laboratory (https://landscan.ornl.gov/) | Yes |
| WorldPop | 2000–2020 | 3 arc-s (~100 m) | Tatem[18] | University of Southampton (https://www.worldpop.org/) | Yes |
| HYDE | 10000 BC–2000 | 5 arc-min (~ 10 km) | Klein Goldewijk et al.[19] | University of Utrecht (https://landuse.sites.uu.nl/datasets/) | No (too low spatial resolution) |
| HRSL (Meta) | 2015 | 1 arc-s (~30 m) | Tiecke et al.[20] | CIESIN (https://www.ciesin.columbia.edu/data/hrsl/) | No (misses China and Russia; no reference years in 1975–2010) |
| Kontur | 2015 | 400 m (Hex. Grid) | Kontur Inc.[21] | Humanitarian Data Exchange (https://data.humdata.org/dataset/kontur-population-dataset) | No (no reference years in 1975–2010) |

Reference years are available in 5-year intervals, except for WorldPop (yearly grids). Due to the absence of validation data for more recent years, only data up to 2010 are analysed

significant knowledge gap in the assessment of their suitability for applications exceeding the urban domain.

This paper addresses this knowledge gap and systematically evaluates the accuracy of global gridded population datasets in rural areas across the planet. As ground-truth data, we employ a combination of reported human resettlement numbers[27] and reservoir surface polygons[28] from 307 large dam construction projects spread over 35 countries (Fig. 2). The resettlement numbers[27] were reported by national dam authorities and mostly stem from comprehensive on-the-ground impact assessments carried out during the planning and construction phase of the dam projects[29]. The reservoir polygons[28], usually derived from satellite imagery, represent the areas inundated upon completion of the dams and thus provide the spatial extents from which the reported number of people[27] were displaced. For further details on the employed data and the methods used for comparing the

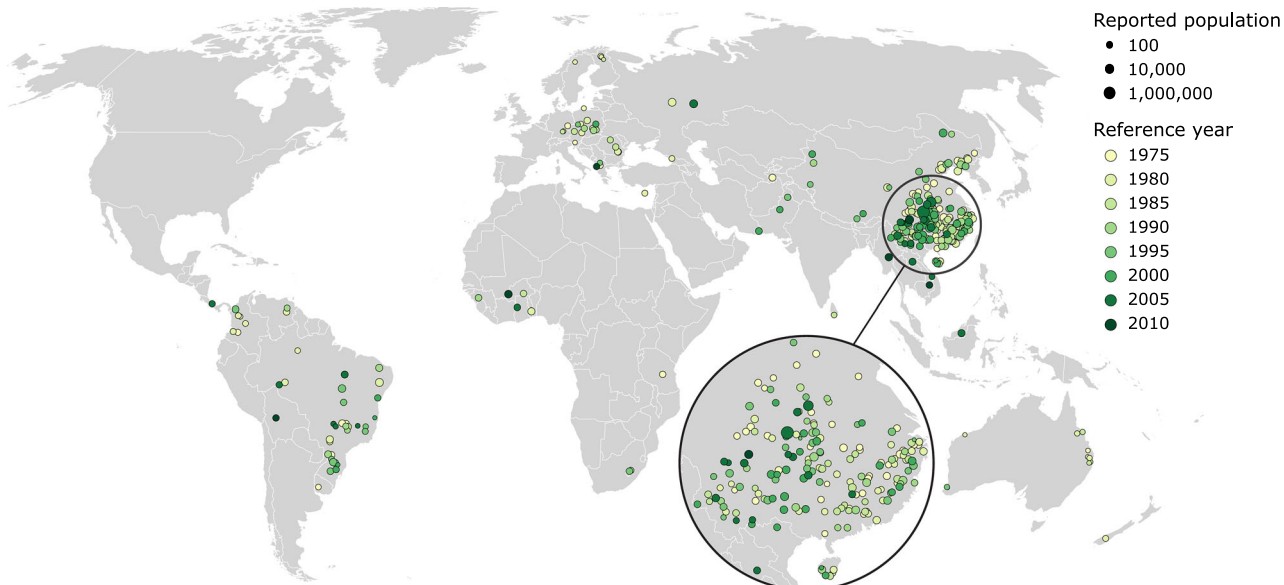

**Fig. 2 | Locations of the 307 rural areas analysed in this study.** The reported population numbers are indicated by marker size, while reference years of the rural areas are shown by marker colour. Country boundaries courtesy of ©EuroGeographics.

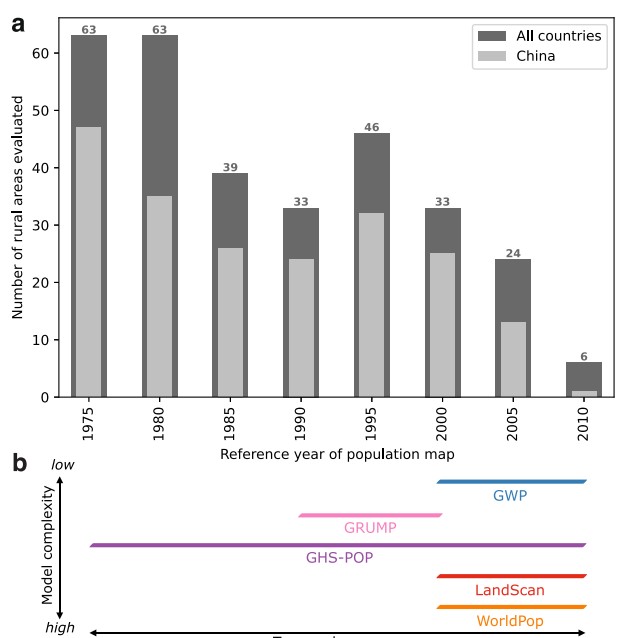

**Fig. 3 | Characteristics of evaluated rural areas and population datasets.**
**a** Temporal distribution of the 307 evaluated rural areas. **b** Temporal coverage and model complexity of the five population datasets examined in this study.

ground-truth data to the gridded population datasets, please refer to section "Methods".

One major advantage of using a combination of resettlement data and reservoir polygons is that the evaluated spatial units relate to the areas covered by water reservoirs, rather than administrative boundaries used for census collection and the creation of the population datasets. To the best of our knowledge, global gridded population datasets have never before been validated using multi-national reference data that are fully independent from population censuses. The reference data contains historical resettlement numbers over the

past decades, which enabled us to carry out a multi-temporal accuracy assessment across the population map reference years 1975–2010 (Fig. 3).

We show that all five examined population datasets systematically underrepresent rural population, with substantial negative biases ranging from −53% (WorldPop) to −84% (GHS-POP). The biases we identify call for a critical discussion of past and future applications of these datasets, to mitigate the risk of rural populations experiencing systematic disadvantages in the allocation of resources and services. Moreover, given that national population censuses are the key input for population models, our results suggest that the incompleteness of censuses in rural areas is a more serious issue than previously acknowledged.

## Results
This section presents the accuracy assessment of five global gridded population datasets in rural areas: GWP, GRUMP, GHS-POP, LandScan, and WordPop. First, we identify systematic differences between the population grids based on the 33 rural areas covered by all five datasets. Next, we break down the results into map reference year and country income level, which are two aspects commonly suspected to influence map accuracy[12]. Finally, we present bias percentages for the five datasets in each of the 35 countries.

### Systematic differences between the gridded population datasets
The year 2000 is the only reference year present in all five examined population datasets (Fig. 3). Therefore, this reference year presents the unique opportunity to inspect systematic differences between the datasets, isolated from temporal influences. Figure 4 contrasts the reported people resettled from reservoir polygons $P_{reported}$ with populations predicted in these areas by the five gridded population datasets $P_{predicted}$ (see section "Methods"). The great majority of predictions across the five datasets significantly underestimate reported values with reference year 2000, whereas overestimations are mostly limited to the data points with less than ten reported people. When visually comparing the results for the five grids, large discrepancies become apparent, with estimates for the same rural area often spanning several orders of magnitude (Fig. 4). The differences between GWP, GRUMP, LandScan, and WorldPop are large but mostly lie within

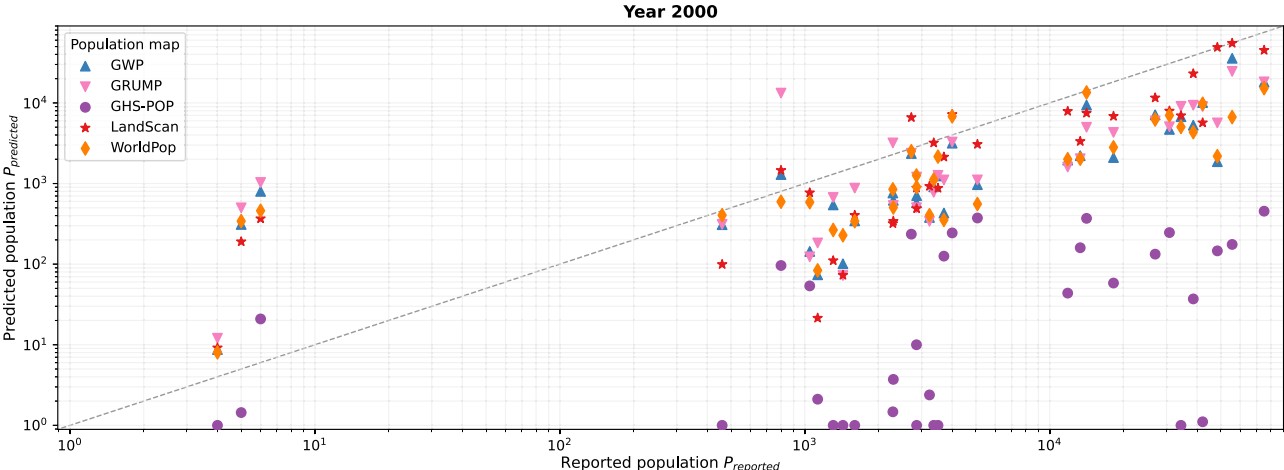

**Fig. 4 | Comparison of reported rural populations (x-axis) with those predicted by the five population datasets (y-axis) for the reference year 2000.** Each of the vertically aligned groups of five data points represents one of the 33 rural areas evaluated for the reference year 2000. Note the logarithmic scale on both axes.

the same order of magnitude. GHS-POP, however, diverges even more substantially from the others: it shows the lowest estimate for all 33 rural areas and consistently underestimates reported population. Further, it appears that GHS-POP has significant gaps in rural areas, estimating values close to zero for several areas with more than 1000 reported people.

### Results by reference year

Figure 5a–e expand the analysis to the other reference years and reveals a systematic and significant negative bias in all five datasets, ranging from −53.4% (WorldPop), −65.0% (GWP), −66.9% (GRUMP), and −68.4% (LandScan) to −83.8% (GHS-POP). This means that even for the data source with the least bias (WorldPop), the predicted rural population amounts to less than half the reported figures. In terms of error variability, the results across data sources are very similar, with a symmetric mean absolute percentage error (sMAPE; see section "Methods") of 0.52–0.53, except for GHS-POP showing significantly higher spread than the others (sMAPE = 0.87). Putting these results into context with the underlying model complexities (Fig. 3b), it appears that more complex population models do not necessarily result in higher accuracy in rural areas.

While a strong negative bias and a high error variability are present across all reference years, they tend to become less evident for more recent years (Fig. 6a, b). This is sensible due to strengthened population census efforts as well as the increasing availability of ancillary data over time[12,13]. While no significant bias trends are visible for GWP, GRUMP, GHS-POP, and LandScan, the improvement of WorldPop is remarkable, having reduced its bias from about −80% to −32% in the years 2000 and 2010, respectively (Fig. 6a). Error variability has improved considerably, with the year 2010 showing the lowest sMAPE for all five datasets (Fig. 6b).

### Results by country income level

Population modellers commonly suspect their datasets to be more accurate in higher-income countries due to more frequent and reliable censuses and greater availability of ancillary data[12], and GHS-POP has indeed shown such a behaviour in urban areas[25]. Figure 5f–j shows our validation results categorised by country income level of the World Bank classification[30]. Among the 307 rural areas, only 22 are located in high-income countries, all with reference years before 2000. The large majority of analysed rural areas lie in low or lower-middle income countries. In Fig. 5f–j, no clear effect of country income on the accuracies of the five datasets can be observed. This initial impression is corroborated by the trend analysis presented in Fig. 6c, d, in which

we focus on the 63 rural areas with reference years 2000–2010 to enable an analysis independent from variations in data accuracy over time (as identified in Fig. 6a, b). The trend analysis confirms that country income level does not significantly influence the accuracy of population datasets in rural areas (Fig. 6c, d).

### Results by country

Finally, we present the results separately for the 35 countries with evaluated rural areas. The global map of bias percentages in Fig. 7 reveals that also at country level, gridded population datasets largely underestimate rural population. Relatively accurate estimates are limited to Austria, Lesotho, Cambodia, and Vietnam (Fig. 7), but these countries are represented by only 1–2 data points (Fig. 8) and thus results are highly uncertain. More reliable conclusions can be drawn for countries with higher numbers of data points, especially for China (203), Brazil (30), Australia (10), Poland (9), and Colombia (6). In all of these countries, rural population is significantly underestimated by all five datasets (Figs. 7 and 8). Positive biases can be observed only in seven countries with few data points (i.e. Venezuela, Ghana, Albania, North Macedonia, Lesotho, Cambodia, and Vietnam), and in none of these the overestimation is unanimous among all five datasets (Fig. 8).

GHS-POP shows great underestimation in all countries except for Austria, North Macedonia, and Cambodia. The other four datasets show a slightly more balanced behaviour. WorldPop appears to have the least bias in many countries, especially in those with higher numbers of data points (e.g. in China, Poland, and Pakistan). In Brazil, however, WorldPop is outperformed by GWP and GRUMP. For a complete list of the numeric bias scores and evaluated map years in each country, please refer to Supplementary Table 1.

Many countries are represented by only a few data points, which impedes a reliable statement on the general accuracy of the maps in their rural areas. Thus, our results in these countries contain significant uncertainties and do not necessarily provide a representative view of the accuracy of the data at national level. However, they provide first indications on the accuracy that should be substantiated with further national-scale accuracy assessments.

## Discussion

**Main implications of the results.** The findings from this study hold significant implications for a wide array of research and policy fields that consider rural areas and their populations, including disaster preparedness, public health planning, environmental conservation, and, ultimately, sustainable development. We assessed the accuracy of global gridded population datasets specifically in rural areas around

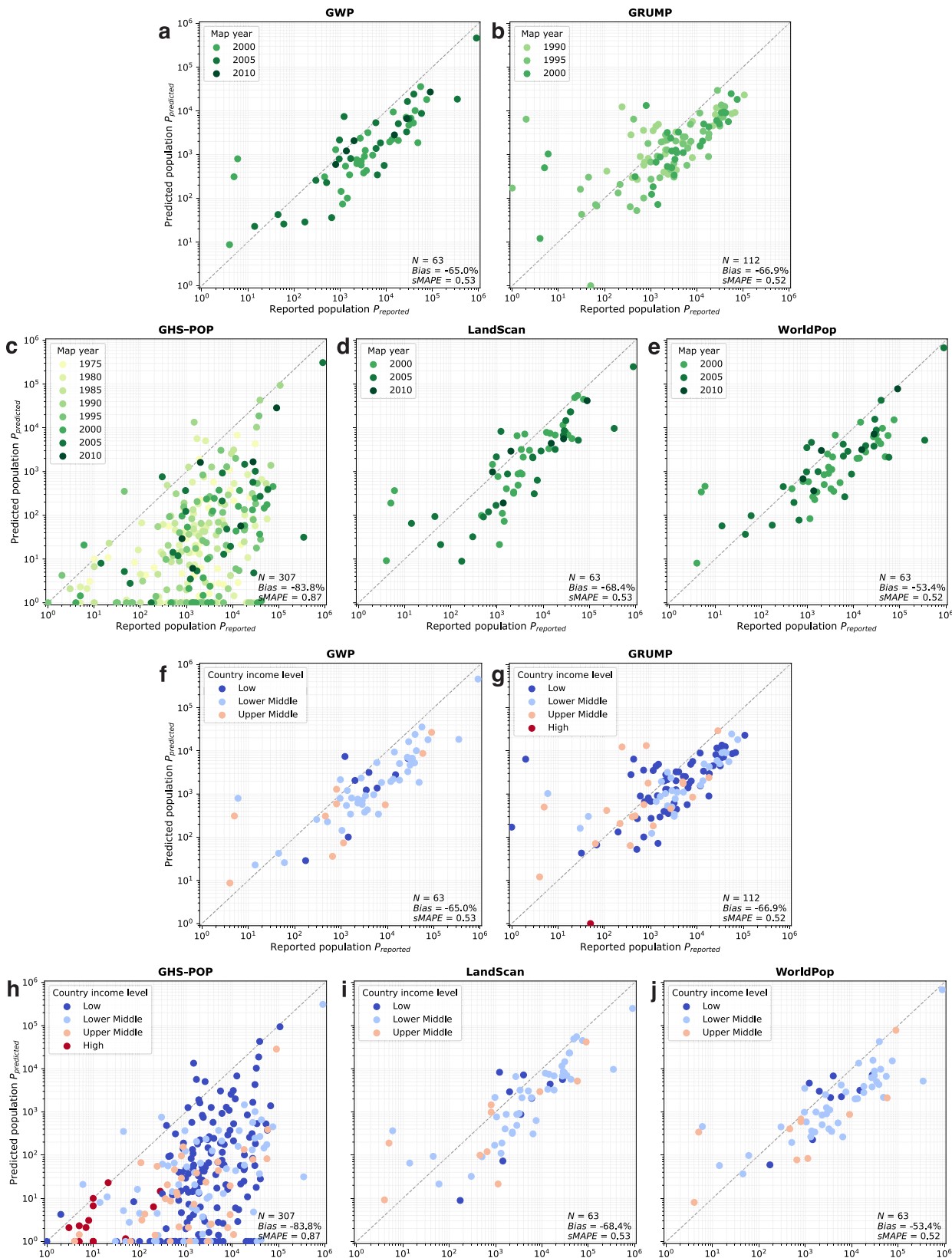

**Fig. 5 | Validation scatter plots for the five analysed population datasets, comparing reported rural populations (*x*-axes) with those predicted by the population datasets (*y*-axes).** Each data point represents one of the 307 analysed rural areas with colouring according to reference year (**a**–**e**) and colouring according to country income level using World Bank classification (**f**–**j**). The accuracy metrics shown in all plots represent bias percentage and symmetric mean absolute percentage error (sMAPE). Note the logarithmic scale on all axes.

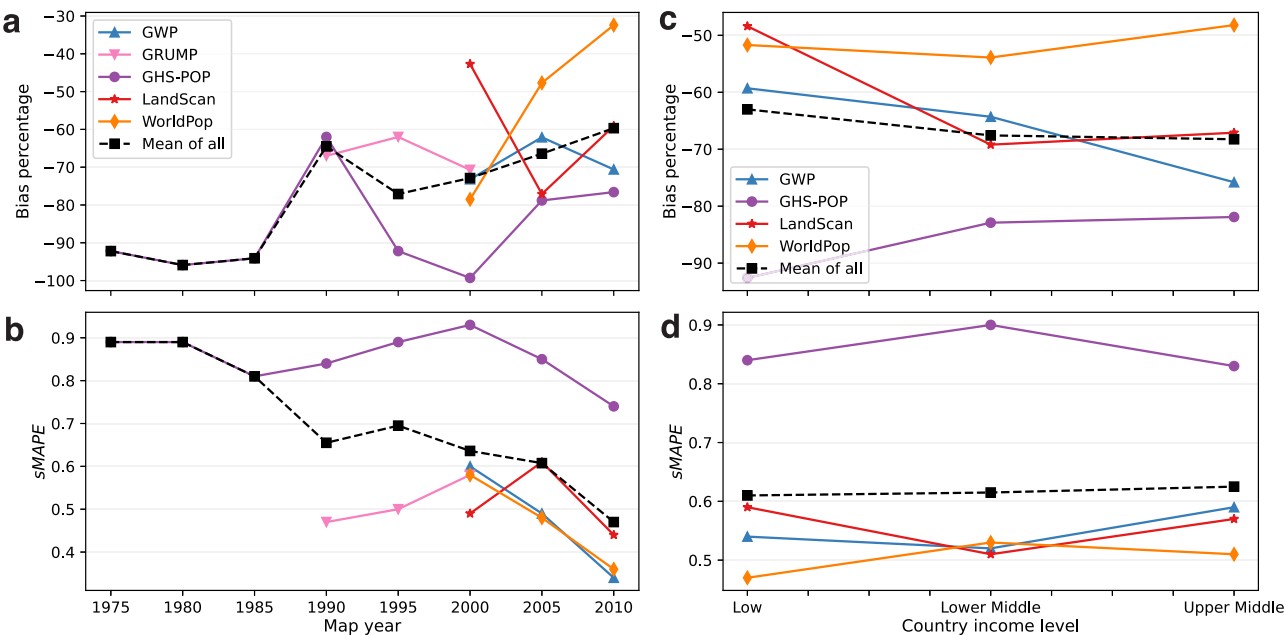

**Fig. 6 | Trend analyses of dataset accuracy over map reference year and country income level. a, b** Influence of map reference year on dataset accuracy. **c, d** Influence of country income level on dataset accuracy. The analysis in **c**, **d** is based solely on the 63 areas with reference years 2000–2010 to exclude effects of different time periods covered by the different population datasets. The accuracy metrics used are bias percentage and symmetric mean absolute percentage error (sMAPE); for both accuracy metrics, the optimal value is zero.

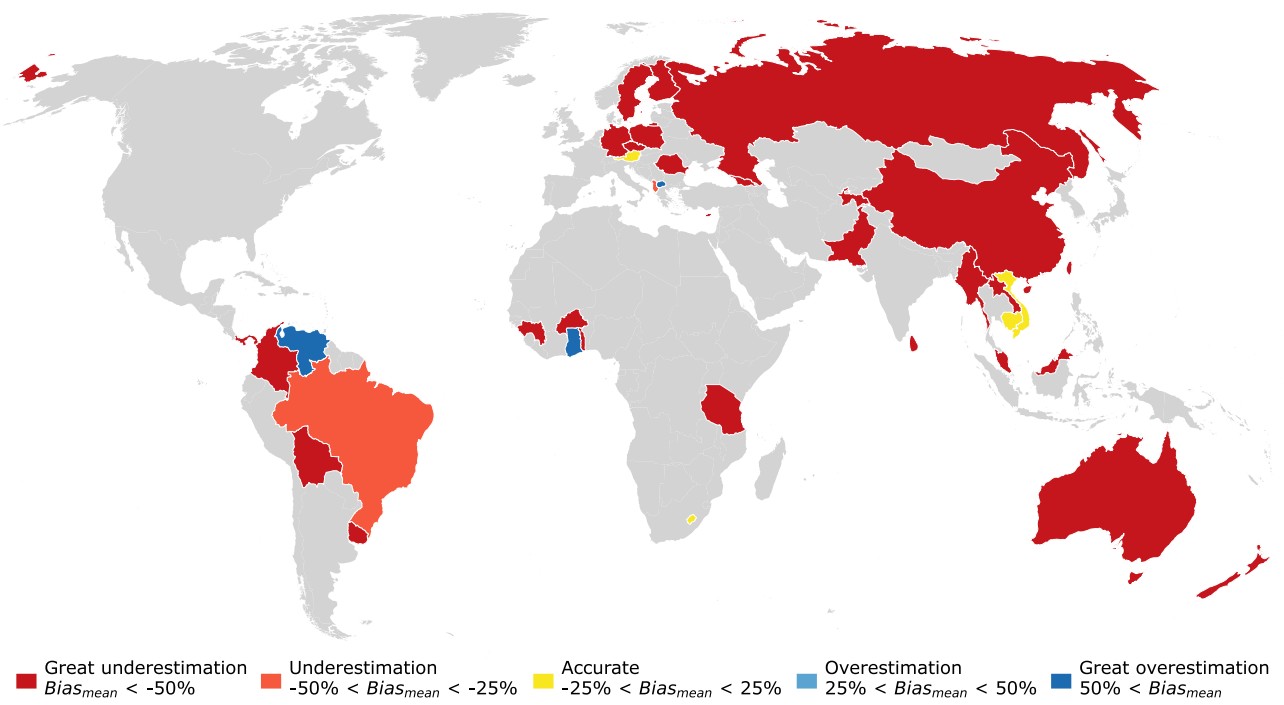

| Great underestimation | Underestimation | Accurate | Overestimation | Great overestimation |
|---|---|---|---|---|
| $Bias_{mean} < -50\%$ | $-50\% < Bias_{mean} < -25\%$ | $-25\% < Bias_{mean} < 25\%$ | $25\% < Bias_{mean} < 50\%$ | $50\% < Bias_{mean}$ |

**Fig. 7 | Mean bias percentages over the five population grids in the 35 countries with evaluated rural areas.** Note that most countries include data points for only some of the five datasets, as indicated in Fig. 8. Country boundaries courtesy of ©EuroGeographics.

the globe using reported human resettlement numbers from over 300 dam projects, which provide multi-national reference data fully independent from population censuses. We found a significant and systematic tendency for all datasets to underestimate rural population, with biases ranging from −53% (WorldPop) to −85% (GHS-POP). This is remarkable, as countless studies have employed these datasets without questioning their accuracy in the rural domain, and the systematic

underrepresentation of rural population directly propagated into their results. It implies that the results of such studies, especially those focusing on rural applications, unknowingly underrepresented the interests of rural populations. For instance, studies that map the potential impacts of disasters on population[9,10,31–33] have likely underestimated the population exposed in rural areas, which may result in an unequitable distribution of risk reduction efforts favouring urban and

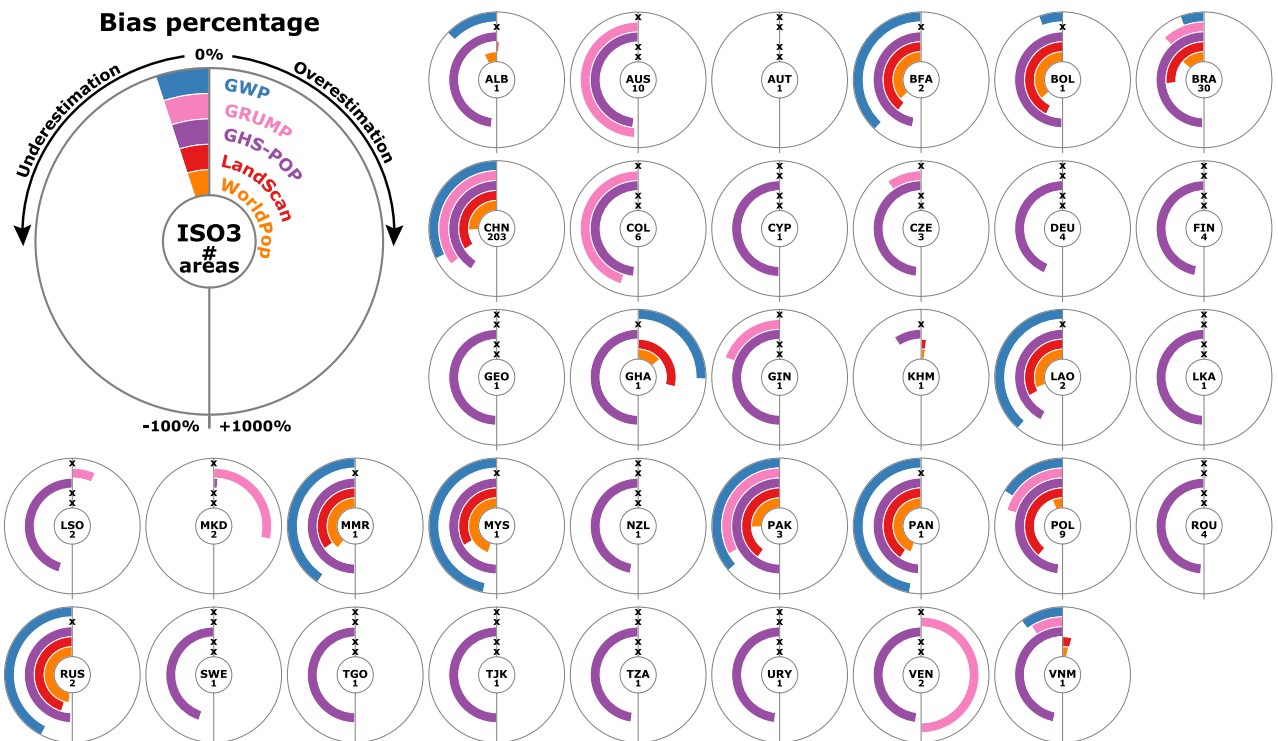

**Fig. 8 | Bias percentages for each of the five population grids in the 35 countries with evaluated rural areas.** Countries are sorted alphabetically by ISO3 country code. The numbers below the ISO3 country codes indicate the totals of rural areas evaluated for each country. The lack of reference years for computing bias percentage of a given dataset is indicated by x-symbols.

discriminating rural population. Or, past analyses of healthcare accessibility[3,4] may have guided policy makers to an insufficient development of healthcare services in rural areas, simply because the real demand of the rural population was not adequately reflected in the data. Policies that build upon such studies have likely been causing population in rural regions, currently forming about 43% of global population[34], to experience systematic disadvantages in accessing services, resources, and equal opportunities for development. To ensure that rural population is not left further behind, past and future studies employing these datasets must undergo a critical discussion of the underlying uncertainties and limitations, encouraging policy makers to a more careful interpretation of the studies' results particularly in rural areas. Otherwise, the fundamental objective of the 2030 Agenda for Sustainable Development to "leave no one behind"[35] will remain an unfulfilled promise.

**Root causes of systematic underrepresentation of rural population.** Since the systematic underrepresentation of rural population is present in all five examined datasets, it seems likely that its root cause lies not primarily in the methods and modelling approaches behind the datasets, but in the underlying input data. The most important input for all population models are national population censuses[12], which are subject to fundamental limitations[13]: Most importantly, insufficient financial resources compromise the completeness and timeliness of the censuses. Furthermore, communities in remote locations or impacted by conflict and violence are difficult to access, and census enumerators often face language barriers and resistance to participation. Such challenges can lead to a substantial incompleteness of the census. For instance, in Paraguay, the 2012 census may have missed a quarter of the population[13]. The significant underrepresentation of rural population we found in the gridded datasets raises the question whether such gaps in the census are more widespread than assumed to date, and how reliable current global

population estimates really are. For example, is it possible that global population estimates from the United Nations[36] (7.98 billion in 2022) or World Bank[37] (7.95 billion in 2022), both relying heavily on national population censuses, miss a significant part of the world population?

Further, the question arises whether gaps in the census affect the accuracy of the gridded population datasets equally in urban and rural areas. Kuffer et al.[25] assessed the accuracy of GHS-POP in selected urban areas around the globe and found high uncertainties but no systematic underestimation. This indicates that, at least for this dataset, the incompleteness in the census does not primarily affect accuracy in the urban domain but it is disproportionally allocated to rural areas. In fact, also the producers of the datasets acknowledge that existing gridded population datasets were mainly calibrated for the urban domain, since this is where the majority of people live, where census data are often available in higher spatial and temporal resolution, and where infrastructure is easier to detect[12]. For rural areas, censuses are subject to more pronounced limitations, and villages and infrastructure may be harder to identify using remote sensing data due to dense tree coverage or differing building types and materials[12]. The use of satellite-based night-time lights by GRUMP and WorldPop[12] may introduce further biases, since rural areas, especially in developing countries, may lack electrification and thus make residential buildings appear uninhabited[38]. Looking at our validation dataset, however, we did not observe any significant influence of country electrification on population estimation bias of GRUMP and WorldPop in rural areas (see Supplementary Fig. 2).

Another potential cause for systematic biases is the spatial resolution of ancillary data: Satellite imagery at 100 m resolution used by GHS-POP and WorldPop may be well suited to delineate urban areas but are too coarse to detect scattered hamlets and villages[13]. For instance, GHS-POP uses a rigid mask of satellite-based building footprints (Fig. 1) that has been shown to detect only about 4% of buildings

in rural areas[20], and this likely caused it to have the largest biases among the five datasets. The recent HRSL population dataset[20] presents an interesting alternative as it uses remote sensing data in 30 m resolution (Table 1), but it currently includes only about 190 countries and lacks several highly populated nations such as China and Russia. Besides, in spite of its high ability to detect rural settlements (e.g. 83% of rural buildings in a case study in Malawi[20]), HRSL has shown even lower rural population estimates than LandScan and WorldPop in a recent large-scale study[31], indicating an even more pronounced negative bias in rural areas than these two datasets we examined.

Finally, it is important to mention that dominant root causes of rural population underrepresentation may vary significantly among countries since the quality of the census and ancillary datasets is inhomogeneous. Further comparative studies in countries where census data are relatively robust would help improve our understanding of the strengths and limitations of the different population models in various contexts.

**Recommendations for mitigating negative biases.** The results of this study call for an improved population data collection and calibration of population models in the rural domain. It is evident that more resources should be allocated to population census efforts specifically in rural areas. In addition, we argue that data producers should employ also alternative population counts to enrich the data collected from censuses and improve model calibration in the rural domain. Alternative population counts include for instance representative household surveys in selected rural areas, such as demonstrated by Boo et al.[39] in the Democratic Republic of Congo, but also reported resettlement from surface mining or large infrastructure projects, such as the data used for validation in this study. Ancillary data sources with limitations in rural areas (such as building footprints or night-time lights), though containing very useful information for the urban domain, should be treated with care and assigned with reasonably low influence in the population models to avoid systematic penalisation of rural population. The POPGRID data collaborative (https://www.popgrid.org) presents an excellent platform for sharing advanced calibration and validation methods for the rural domain among the community of population modellers.

Several past applications of gridded population datasets have shown that study results, particularly in sparsely populated regions, are highly sensitive to the population dataset choice[3,6,31], and the large discrepancies we found between the examined datasets corroborate these previous findings. However, most studies use only one of the available population datasets and do not critically discuss their choice; the decision to use a particular dataset is often driven by ease of access and use, rather than by the suitability of the dataset for the given study context[3,12,22]. Our comparative analysis directly supports users in their dataset choice, especially for studies involving rural areas: We recommend the use of WorldPop for global and large-scale analyses due to the least pronounced (yet remarkable) systematic bias found among the five datasets. WorldPop has been endorsed also by previous analyses in regions containing urban areas[22–24]. For smaller-scale studies focusing on rural areas, the country-specific bias scores we provide in Fig. 8 and in Supplementary Table 1 offer quantitative guidance on the most accurate dataset in the region of interest. To fully support an informed dataset choice, population modellers themselves could routinely assess their products and openly provide reliability estimates for their data in both rural and urban domains.

**Limitations of the study.** The results of this study are subject to uncertainties, primarily concerning the resettlement numbers reported by national dam authorities and used for validation (see Methods). These numbers may at times represent informed estimates rather than exact ground-based population counts. Unfortunately, no information was available on potential biases in the resettlement data. One possible source of bias could be that national dam authorities tend to underreport resettlement to downplay the social impact of dams[40]. In the

context of our analysis, such underreporting would further amplify the negative biases we identified for the gridded population datasets.

Secondly, the simple bias adjustment we applied to the predicted population values to mitigate underrepresentation of reservoir areas in the GeoDAR dataset (see Methods) affects our assessment of error variability, since some reservoir areas are more accurately represented than others. This artificially increases the error variability results presented as sMAPE scores. However, the bias percentages that form the main results of this study are not affected and should be robust given the high number of evaluated rural areas ($N = 307$).

Thirdly, the lack of validation data for the years after 2010 impeded an accuracy assessment of population grids for the years 2015 and 2020. Our analysis showed the least mean bias for the latest evaluated reference year 2010, and the accuracy of the datasets may have further improved for more recent years, for instance due to the increased implementation of population registers and register-supported censuses adopted by many countries[41]. This would imply less significant biases than those we identified here for the period 1975–2010.

Lastly, although the analysed set of 307 rural areas shows large variety (Supplementary Table 2), it is not a representative sample in statistical terms that would allow us to evaluate population datasets for the whole global rural population. Nevertheless, our results provide a clear indication that rural populations tend to be underestimated by global population datasets. To further corroborate our findings for rural areas as a whole, we recommend additional validation studies using reference data from other contexts (e.g. resettlement data from surface mining activities).

## Methods
### Global gridded population datasets
From the eight existing gridded population datasets with (near-)global coverage listed in Table 1, five datasets were selected that contained map reference years within the temporal coverage of the validation data (1975–2010): GWP, GRUMP, GHS-POP, LandScan, and WorldPop. All these products provide maps in five-year intervals (except for WorldPop providing yearly grids), and this temporal resolution has thus been adopted for the analysis.

For each of the five datasets, the highest available spatial resolution has been selected for the main analysis, i.e. 1 km for GWP, GRUMP, and LandScan, and 100 m for GHS-POP and WorldPop (results for WorldPop in 1 km resolution can be found in Supplementary Fig. 3). For WorldPop, the version unconstrained by land use has been selected due to the unavailability of the constrained version for reference years before 2020. For those products that offer versions with and without adjustment using national-level estimates from United Nations Population Division (i.e. GWP, GRUMP, and WorldPop), both versions have been included in the initial analysis. In this paper, however, we show only the results for the UN-adjusted grids since these performed slightly better than their unadjusted counterparts (Supplementary Fig. 4 shows results for the unadjusted datasets).

### Validation data
The numbers of people resettled due to large dam projects are provided by the International Commission on Large Dams (ICOLD). The ICOLD World Register of Dams[27] is a continuously updated database of currently about 62,000 dams, including various dam and reservoir attributes reported by the national dam authorities of 106 member countries. In this study, three reservoir attributes have been used: The number of resettled people, the maximum reservoir surface area in km$^2$, and the year of dam completion. Only 2699 reservoirs in the database (about 4% of entries) contained resettlement data, with overall 10.02 million displaced people. The resettlement numbers typically stem from comprehensive on-the-ground surveys as part of impact assessments carried out during the planning and authorisation

procedure of dam construction projects[29]. An inquiry we sent to ICOLD Central Office confirmed that the numbers represent only physical resettlements from areas that were later inundated and occupied by reservoirs and do not include secondary displacements of people residing outside the reservoir areas, e.g., due to livelihood loss induced by the project (personal communication, 28.8.2023). For further considerations on uncertainties in the resettlement numbers, see section "Discussion".

While the ICOLD database covers human resettlements and other numeric attributes of reservoirs, it does not contain information on the reservoirs' geographic coordinates and spatial extents. To bridge this gap, Wang et al.[28] developed a method for geo-referencing reservoirs in the ICOLD database. They employed geocoding techniques to connect the entries in the ICOLD database to three publicly available global maps of water bodies and reservoirs, namely GRanD[42], HydroLAKES[43], and UCLA Circa 2015 Lake Inventory[44]. The resulting reservoir polygons spatially resolved over 90% of global reservoir surface area, which makes GeoDAR to date the most comprehensive data source that includes both reservoir attributes and their geographic information[28].

We updated the reservoir attributes in the GeoDAR data with the most recent information from the ICOLD database (retrieved on 20 July 2023). A subset of reservoirs has been selected that fulfils the following minimum data requirements for the analysis: (i) resettlement data available, (ii) reservoir surface area larger than 1 km², (iii) construction completed earliest in 1980, and iv) population density in the reservoir area below 1500 people/km², a threshold used by World Bank[45] and United Nations Statistical Commission[46] to delineate cities based on population density. To avoid ambiguities, we manually removed transboundary dams and those with ICOLD attributes indicating a previously existing reservoir in the same location (e.g. dam reconstructions or heightenings). This resulted in a final set of 307 reservoirs for analysis in this study with a combined surface area of 22,489 km².

Among the 307 reservoirs, relatively small surface areas (1–25 km²) are most common, but also numerous larger areas up to about 4000 km² in size are included. Supplementary Fig. 6 shows the size distribution of the reservoirs and illustrates that area size does not have an influence on the mean bias of the population datasets. In addition to area size distribution, we also analysed the distributions of population numbers, population densities, and altitudes of the 307 areas (Supplementary Table 2). All four sample characteristics include a wide range of values, with standard deviations being larger than means and medians. This large spread in the data implies that our sample of 307 areas covers a great variety of contexts.

The reservoirs are distributed over 35 countries on all continents except for North America (Fig. 2). China is strongly over-represented with 203 reservoirs, but due to many small reservoirs these cover only about 26% of the combined surface area of all included reservoirs. Nonetheless, the validity of the results for the global domain has been confirmed by carrying out a second analysis without Chinese reservoirs that yielded very similar results (see Supplementary Fig. 5).

**Selection of the population map reference years**
The year of dam completion in the ICOLD database represents the situation when a reservoir has been filled and people have already been relocated from the area. To find for each reservoir the population map year that corresponds to the situation before resettlement, the year of dam completion requires a temporal offset. According to Ansar et al.[47] the construction of very large dams takes on average 8.6 years, and the resettlement of people is typically a gradual process over the construction period. To obtain for each rural area a map reference year from the 5-year intervals within 1975–2010, we therefore chose a temporal offset of 5–9 years before dam completion as reference year

$Y_{reference}$ (Eq. 1). In other words, the completion year $Y_{damcompletion}$ of each dam i is rounded down to the second closest population map reference year. A sensitivity analysis supported the robustness of this choice (see Supplementary Fig. 7 for results assuming an alternative temporal offset of 10–14 years). The resulting distribution of the 307 reservoirs over the reference years within 1975–2010 is shown in Fig. 3.

$$Y_{reference, i} = Y_{damcompletion, i} - (5 + Y_{damcompletion, i} \bmod 5) \qquad (1)$$

**Prediction of rural population**
Following these preparations, we retrieved from the gridded population datasets the number of people located inside the 307 reservoir areas. This was done by a simple spatial overlay of each of the reservoir polygons with the population grids of the according reference year $Y_{reference}$: First, the spatial resolution of the population grids was increased to 10 m to mitigate data artefacts related to coarser cell sizes. This was done by taking the population totals at the native resolution and evenly distributing them among the high-resolution cells. Then, the values of all high-resolution pixels with centroids located inside the reservoir polygon were summed to determine the total number of people $P_{polygon}$.

The assumed even population distribution within low-resolution cells may introduce uncertainties, as in reality, populations may be concentrated in small parts of the cells. Therefore, we tested the sensitivity of assuming an even population distribution by comparing the bias results for WorldPop using 100 m resolution data (as in the main analysis) against the results using the aggregated 1 km data (see Supplementary Fig. 3). This resulted in marginal differences relating likely to the Modifiable Areal Unit Problem (MAUP)[48] rather than uneven population distributions inside the cells.

As pointed out in previous research[49], the reservoir polygons in the GeoDAR dataset tend to underestimate the real reservoir areas since the polygons are usually derived from satellite images that may show a situation when the reservoir is not fully filled and thus not at its maximum surface extent. To mitigate the effect of this systematic area underestimation, the rural populations estimated above ($P_{polygon}$) require a bias adjustment. We first quantified the systematic underrepresentation by comparing surface areas reported by ICOLD to the corresponding geometric areas of the GeoDAR polygons (Fig. 9). This analysis included 6095 reservoirs with reported surface areas larger than 1 km², after 209 outliers were removed due to unrealistically large

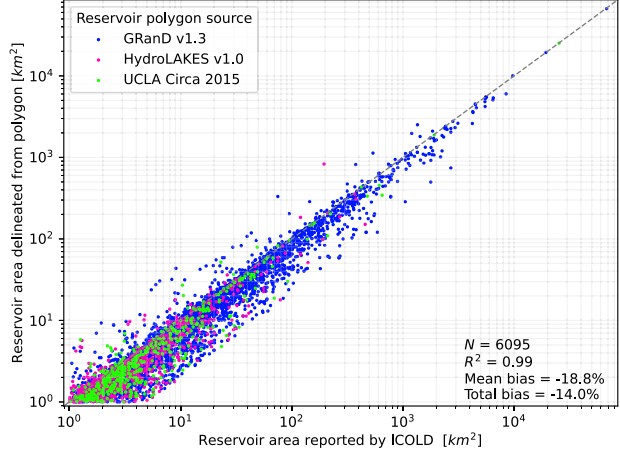

**Fig. 9 | Surface area validation of three reservoir polygon sources combined in GeoDAR (i.e. GRanD, HydroLAKES, and UCLA Circa 2015) against surface areas reported by ICOLD.** The validation shows a systematic underrepresentation of the real reservoir area by GeoDAR polygons (mean bias = –18.8%).

differences between the two area values (ratio > 5). We found that GeoDAR polygons underrepresent the real reservoir extent by on average 18.8%.

Second, we used the identified mean bias of −18.8% to adjust the numbers of people retrieved from the population maps for each reservoir polygon $i$ ($P_{\text{polygon},i}$), resulting in final predicted population amounts $P_{\text{predicted},i}$ (Eq. 2). This bias adjustment assumes that population density in the missing part of each reservoir area is equal to population density inside the reservoir polygon.

$$P_{\text{predicted},i} = \frac{1}{1 - 0.188} \cdot P_{\text{polygon},i} = 1.23 \cdot P_{\text{polygon},i} \qquad (2)$$

### Accuracy evaluation

To analyse the accuracy of the population grids in rural areas, we compare the predicted population numbers ($P_{\text{predicted}}$) to those reported by ICOLD ($P_{\text{reported}}$). This comparison is carried out by means of two accuracy metrics:

First, bias percentage is used to analyse the general over- or underestimation of rural population (Eq. 3). Positive or negative values imply systematic over- or underestimation, respectively. A value of Bias = 0% represents the absence of biases.

$$\text{Bias} = \frac{\sum P_{\text{predicted},i} - \sum P_{\text{reported},i}}{\sum P_{\text{reported},i}} \cdot 100\% \in [-100\%, +\infty] \qquad (3)$$

Second, symmetric mean absolute percentage error (sMAPE) has been chosen to detect error variability of the data points (Eq. 4). While a value of sMAPE = 0 means a perfect prediction without deviations from reported values, sMAPE = 1 implies an average estimation error of 100%.

$$\text{sMAPE} = \frac{1}{n} \sum_{i=0}^{n} \frac{\left| P_{\text{reported},i} - P_{\text{predicted},i} \right|}{\left| P_{\text{reported},i} \right| + \left| P_{\text{predicted},i} \right|} \in [0, 1] \qquad (4)$$

### Reporting summary

Further information on research design is available in the Nature Portfolio Reporting Summary linked to this article.

## Data availability

The five gridded population datasets are freely accessible through the weblinks provided in Table 1. The GeoDAR dataset[28] containing the reservoir polygons is freely available for download on Zenodo (https://zenodo.org/records/6163413). The reservoir attributes need to be purchased from the International Commission of Large Dams through the World Register of Dams[27] (https://www.icold-cigb.org/GB/world_register/world_register_of_dams.asp). The polygon file resulting from this study is publicly available (https://doi.org/10.5281/zenodo.14637154)[50], and it contains the 307 evaluated rural areas and the population estimates for each rural area.

## Code availability

The Python code for generating the population estimates for the 307 rural areas is openly accessible on Zenodo (https://doi.org/10.5281/zenodo.14637154)[50] and GitHub (https://github.com/josiasritter/population_grid_assessment). The repositories also contain code for the validation procedure and for generating the figures in Results.

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

## Acknowledgements

We would like to express our gratitude to the International Commission for Large Dams (ICOLD) for providing the dam attribute data, and to Jida Wang for providing the GeoDAR dataset and kind instructions on updating the GeoDAR polygons with the most recent ICOLD data records. Many thanks also to Marko Kallio, Matti Kummu, Olli Varis, and Venla Niva for fruitful discussions. We would like to acknowledge the Department of Built Environment at Aalto University for funding the main part of this study as well as the Digital Waters Flagship (decision no. 359248) funded by the Research Council of Finland. Lastly, the biggest thanks are owed to the developers of the population datasets evaluated in this study for taking on the vital yet extremely challenging task to estimate population distribution at the global scale, and for publishing these datasets with open access.

## Author contributions

J.L.R. conceptualised the study with contributions from M.K. and H.T. J.L.R. developed the software codes for data analysis and visualisation. The authors jointly carried out the formal analysis of the results. J.L.R. prepared the initial manuscript draft, which was subsequently reviewed and edited by M.K. and H.T.

## Competing interests

The authors declare no competing interests.
