## [Transparent Peer Review file · Nature Communications]

Global gridded population datasets systematically underrepresent rural population

Corresponding Author: Dr Josias Láng-Ritter

Version 0:

Reviewer comments:

Reviewer #2

(Remarks to the Author)

This paper provides an assessment of accuracy for five global gridded population datasets in rural areas. The authors compare global gridded datasets (WorldPop, GWP, GRUMP, LandScan, GHS-POP) and human resettlement data from 307 dam construction projects located in 35 countries. This is novel research, and as the authors pointed out in the Introduction and Conclusions, it may have practical implications. The paper is clearly written and explains the motivation behind the project and the practical implications of the obtained results. However, in my opinion, the authors should rethink the structure of the paper before publication.

My comments on this paper are as follows:

- 1) Why does the "Method and data" section follow the Results and Discussion sections? The information provided in this section is necessary to understand the results.
- 2) Lines 74 and 75 state, "First, we identify systematic differences between the population grids based on those data points present in all five datasets." What are "those data points"?
- 3) Section 2.1: At this point, I don't understand how those comparisons are made. Is the ground-truth represented as point data? How are the differences calculated between point data and grid cells? What are the "reported rural populations Reported values" in Figure 4? I found answers to this question in section 4. In my opinion, it should be explained before describing the results, or at least in the Results section the authors should point out that the details about the data are described later (and provide reference to a particular subsection).
- 4) Section 2.4: I am not convinced by this section. The conclusions are drawn based on a few data points, which may affect their reliability. This is, in my opinion, the weakest part of this paper.
- 5) The methods and data section thoroughly discusses the data, preprocessing steps, and the limitations of the analysis and results that stem from data and methods. However, as I mentioned above, I don't understand why section 4 is included after the conclusions and results. Part of this information is required to understand the results and findings.

(Remarks on code availability)

Reviewer #3

(Remarks to the Author)

This paper compares gridded population data in rural areas with an independent data source, i.e., resettlement data from dams. Overall the paper is well written and the results are significant. The methodology seems fine, with clear caveats and limitations signposted. I have a few comments that could be still be addressed:

1. I understand why you would disaggregate the data evenly to a 10m grid because the boundaries of your polygons cross

the 1 km grid cells (or finer resolutions for some products) but what would be the effect if instead you allocated all the population in an overlapping grid cell into the reservoir area. I am trying to understand what the effect of this linear distribution has on your results, particularly if populations are concentrated spatially within a grid cell (even though you don't have this information). It would be an interesting type of sensitivity test to see if this makes a difference.

2. I didn't get a strong feel for the size distribution of the polygons. There is a graph that has areas but it would be better to include this as a standalone piece of information so that we get a better feel for the distribution of these areas. Then it would be good to understand if the size of the area has an influence on the bias.

3. Because some of these data sets rely on night time lights, could you select those countries that have the least electrification and compare those to countries with more electrification to see if there is an effect?

4. I think that the graphs by country are interesting (Figure 10) but I am not sure I would make an informed decision about bias when some countries have very few data points. Hence this caveat could be highlighted.

5. What would really be interesting to understand is the source of the bias by country, i.e., is it because the census data are wrong/underestimated or is it the method? I realize that your paper doesn't answer this but it would help make more informed choices. Some further call for comparative studies where census data are more reliable might be good.

Minor comments

1. line 27 - change degree to degrees

2. Supplementary material - section 1 - change scores to score

(Remarks on code availability)

Reviewer #4

(Remarks to the Author)

What are the noteworthy results?

The study finds significant discrepancies in global gridded population datasets when estimating rural populations living near a selection of global reservoirs. The datasets examined—WorldPop, GPW, GRUMP, LandScan, and GHS-POP—underestimate rural populations in these areas substantially, by 53%, 65%, 67%, 68%, and 84%, respectively.

Will the work be of significance to the field and related fields? How does it compare to the established literature?

The work is significant because it highlights a critical problem in widely-used population datasets. However it does overreach on its conclusions by applying lessons learned to rural areas as a whole rather than to the specific case of highly localized rural settlements consisting of small land areas. The study contributes to the literature by providing a global-scale validation of population datasets specifically in rural areas, which has been lacking.

Does the work support the conclusions and claims, or is additional evidence needed? Is the methodology sound? Does the work meet the expected standards in your field?

The work does support conclusions concerning the more narrow and localized geographic scope of analyzing small settlements near a subset of global reservoirs, however the conclusions and claims of the authors that this causes a problem across rural areas as a whole are not fully substantiated. The scale of the analysis is of great importance here. In order to make sweeping statements and conclusions about the accuracy of these data in rural areas as a whole it would be necessary to analyze other rural contexts.

Is there enough detail provided in the methods for the work to be reproduced?

The study employs multiple datasets for cross-validation, and provides detailed explanations of the methods and adjustments used. There is sufficient detail in the methods section to reproduce the work.

(Remarks on code availability)

Since this was a double blind assessment I did not follow the link to the github which would reveal the authors.

Version 1:

Reviewer comments:

Reviewer #2

(Remarks to the Author)

The authors provide an extensive responses to my comments and made appropriate changes in the paper. I am satisfied by the clarification made in the paper.

(Remarks on code availability)

Reviewer #3

(Remarks to the Author)

You have addressed the comments very comprehensively so thank you very much for this. I have no further comments to the manuscript.

(Remarks on code availability)

Unfortunately this is not an area where I can provide much expertise. I hope the other reviewers are able to demonstrate reproducibility/usability.

Reviewer #4

(Remarks to the Author)

Thank you for your detailed responses to the review comments. The paper is much improved and offers potential for interesting continued research. Best of luck!

(Remarks on code availability)

Thank you for sharing the code and feature class of results.

Dear Reviewers 2, 3, and 4,

We would like to express our sincere gratitude for your constructive comments that helped considerably to improve our manuscript. Please find below the replies to your comments as well as the modifications we have introduced.

Reviewer #2

This paper provides an assessment of accuracy for five global gridded population datasets in rural areas. The authors compare global gridded datasets (WorldPop, GWP, GRUMP, LandScan, GHS-POP) and human resettlement data from 307 dam construction projects located in 35 countries. This is novel research, and as the authors pointed out in the Introduction and Conclusions, it may have practical implications. The paper is clearly written and explains the motivation behind the project and the practical implications of the obtained results. However, in my opinion, the authors should rethink the structure of the paper before publication.

My comments on this paper are as follows:

[C2.1] Why does the “Method and data” section follow the Results and Discussion sections? The information provided in this section is necessary to understand the results.

The submission guidelines for manuscripts submitted to Nature Communications (<https://www.nature.com/ncomms/submit/article>) specify the sequence of manuscript sections as follows: i) Introduction, ii) Results, iii) Discussion, and iv) Methods. However, to address your concern, we have added in several instances in the Introduction and Results sections remarks on the main aspects of methods and data, along with cross-references to the subsections providing more details on the methods and data. For the exact modifications we made, please refer to our reply to comment C2.3.

[C2.2] Lines 74 and 75 state, “First, we identify systematic differences between the population grids based on those data points present in all five datasets.” What are “those data points”?

Thank you for pointing out potential confusion in this sentence. We have adjusted the phrasing to improve clarity. It reads now as follows:

“First, we identify systematic differences between the population grids based on the 33 rural areas covered by all five datasets.”

[C2.3] Section 2.1: At this point, I don’t understand how those comparisons are made. Is the ground-truth represented as point data? How are the differences calculated between point data and grid cells? What are the “reported rural populations Preported values” in Figure 4? I found answers to this question in section 4. In my opinion, it should be explained before describing the results, or at least in the Results section the authors should point out that the details about the data are described later (and provide reference to a particular subsection).

As outlined in our reply to comment C2.1, the paper structure is given by Nature Communications. However, we embrace your suggestion to enhance information on the most important aspects of the validation data and methods at an earlier stage of the manuscript. We implemented this in the following instances:

- Section 1 (Introduction), lines 52-60: We added details on the general characteristics of the validation data, with reference to the Methods and data section. The passage reads now as follows:
“As ground-truth data, we employ a combination of reported human resettlement numbers²⁷ and reservoir surface polygons²⁸ from 307 large dam construction projects spread over 35 countries (Figure 2). The resettlement numbers²⁷ were reported by national dam authorities and mostly stem from comprehensive on-the-ground impact assessments carried out during the planning and construction phase of the dam projects²⁹. The reservoir polygons²⁸, usually derived from satellite imagery, represent the areas inundated upon completion of the dams and thus provide the spatial extents from which the reported number of people²⁷ were displaced. For further details on the employed data and the methods used for comparing the ground-truth data to the gridded population datasets, please refer to Section 4.”
- Section 2.1, lines 91-93. We added clearer instructions on how to read the scatter plots in this paper (Figures 4, 5 and 7) and the data presented in them, with reference to the corresponding Methods subsections:
“Figure 4 contrasts the reported people resettled from reservoir polygons $P_{reported}$ (Section 4.2) with populations predicted in these areas by the five gridded population datasets $P_{predicted}$ (Section 4.1).”
- Section 2.2: We added references to the corresponding Methods subsection describing the accuracy metrics used for evaluating the population data.

[C2.4] Section 2.4: I am not convinced by this section. The conclusions are drawn based on a few data points, which may affect their reliability. This is, in my opinion, the weakest part of this paper.

Thank you for pointing out this caveat that has also been noted by reviewer #3 (see comment C3.4). We have rephrased parts of this section and added a paragraph pointing out the limitations of the country-level results more explicitly than in the previous version of the manuscript, which reads as follows (lines 175-179):

“Many countries are represented by only a few data points, which impedes a reliable statement on the general accuracy of the maps in their rural areas. Thus, our results in these countries contain significant uncertainties and do not necessarily provide a representative view of the accuracy of the data at national level. However, they provide first indications on the accuracy that should be substantiated with further national-scale accuracy assessments.”

[C2.5] The methods and data section thoroughly discusses the data, preprocessing steps, and the limitations of the analysis and results that stem from data and methods. However, as I mentioned above, I don't understand why section 4 is included after the conclusions and results. Part of this information is required to understand the results and findings.

As described in our replies to comment C2.1, and C2.3 in more detail, we have added to the Introduction and Results sections short explanations of the main features of employed data and methods with references to the corresponding subsections.

Reviewer #3

This paper compares gridded population data in rural areas with an independent data source, i.e., resettlement data from dams. Overall the paper is well written and the results are significant. The methodology seems fine, with clear caveats and limitations signposted. I have a few comments that could be still be addressed:

[C3.1] I understand why you would disaggregate the data evenly to a 10m grid because the boundaries of your polygons cross the 1 km grid cells (or finer resolutions for some products) but what would be the effect if instead you allocated all the population in an overlapping grid cell into the reservoir area. I am trying to understand what the effect of this linear distribution has on your results, particularly if populations are concentrated spatially within a grid cell (even though you don't have this information). It would be an interesting type of sensitivity test to see if this makes a difference.

Thank you for this suggestion, which we find very intriguing. Indeed, populations in grid cells crossed by a polygon boundary may in reality be concentrated in the cell portion that falls inside (or outside) the polygon. However, the suggested full counting of all crossed grid cells would result in an unfair comparison, since especially for polygons with elongated shapes, this would enlarge the evaluated area to a multiple of the original polygon area (for an illustration of this effect, see an enlargement of Figure 1 further below in Figure R4). However, we investigated the effects of different data resolutions by running the analysis on WorldPop data in 1 km resolution and compared it to the (already existing) WorldPop results using the 100 m resolution data (Figure R1). The result shows that using WorldPop in 1 km resolution leads to very similar results as using the 100 m data (biases of -50.5% and -53.4%, respectively). Such differences are very common in studies combining gridded and vectorized data and often relate to the Modifiable Areal Unit Problem (MAUP; see e.g. Salmivaara et al., 2015: <https://www.mdpi.com/2073-4441/7/3/898>). We have added Figure R1 in the

Supplement and refer to it in the Methods section (lines 410-416) with the following discussion:

“The assumed even population distribution within low-resolution cells may introduce uncertainties, as in reality, populations may be concentrated in small parts of the cells. Therefore, we tested the sensitivity of assuming an even population distribution by comparing the bias results for WorldPop using 100 m resolution data (as in the main analysis) against the results using the aggregated 1 km data (see Supplementary Figure S3). This resulted in marginal differences relating likely to the Modifiable Areal Unit Problem (MAUP)⁴⁶ rather than the assumption of an even population distribution inside the cells.”

Figure R1. Comparison of results using Worldpop data in 100 m resolution (left; as in the manuscript), and in 1 km resolution (right). No significant differences were found in terms of accuracy scores. This figure has been added as Supplementary Figure S3.

[C3.2] I didn't get a strong feel for the size distribution of the polygons. There is a graph that has areas but it would be better to include this as a standalone piece of information so that we get a better feel for the distribution of these areas. Then it would be good to understand if the size of the area has an influence on the bias.

Thank you for this great suggestion. We have added to the Supplement an illustration of the area size distribution of the polygons along with the biases in the different size groups (see below Figure R2). A reference to this analysis have been added in lines 375-381 of the Methods section, where we write:

“Among the 307 reservoirs, relatively small surface areas (1 – 25 km²) are most common, but also numerous larger areas up to almost 4000 km² in size are included. Supplementary Figure S5 shows the size distribution of the reservoirs and illustrates that area size does not have an influence on the mean bias of the population datasets.”

Figure R2. Size distribution of evaluated areas. The bias percentage shown as the bar colouring represents the mean of the bias percentages of the five population datasets. The extreme negative bias for the bar at the upper end is due to all polygons in this size group having reference years before 1990, for which only the least accurate dataset GHS-POP is available. This figure has been added as Supplementary Figure S5.

[C3.3] Because some of these data sets rely on night time lights, could you select those countries that have the least electrification and compare those to countries with more electrification to see if there is an effect?

We investigated this interesting idea by using global data on historical country electrification (World Bank; <https://data.worldbank.org/indicator/EG.ELC.ACCS.ZS>). We carried out a correlation analysis between i) population estimation bias in each polygon and ii) electrification in the polygon's country at the reference year (Figure R3). This was done for WorldPop and GRUMP, which are the two population datasets using night-time lights in their algorithms. Our results show no significant Spearman correlation for either of the two datasets. This suggests that factors such as Census quality or the use of building footprints strongly dominate the use of nightlights. We added Figure R3 to the Supplement and adjusted the part of the discussion section referring to night-time lights, which reads now as follows (lines 241-245):

"The use of satellite-based night-time lights by GRUMP and WorldPop¹² may introduce further biases, since rural areas, especially in developing countries, may lack electrification and thus make residential buildings appear uninhabited³⁸. Looking at our validation dataset, however, we did not observe any significant influence of country electrification on population estimation bias of GRUMP and WorldPop in rural areas (see Supplementary Figure S2)."

Figure R3. Correlation analysis between country electrification and population prediction bias in individual rural areas, for WorldPop and GRUMP. Electrification data was unavailable for some countries and reference years, resulting in a reduced set of rural areas evaluated here ($N_{WorldPop} = 63$; $N_{GRUMP} = 48$). This figure has been added as Supplementary Figure S2.

[C3.4] I think that the graphs by country are interesting (Figure 10) but I am not sure I would make an informed decision about bias when some countries have very few data points. Hence this caveat could be highlighted.

We have reworked the text in this section to guide the reader to a more realistic interpretation of the national-level results where countries have only a few data points. For a detailed description of the changes we made, please refer to our reply to comment C2.4.

[C3.5] What would really be interesting to understand is the source of the bias by country, i.e., is it because the census data are wrong/underestimated or is it the method? I realize that your paper doesn't answer this but it would help make more informed choices. Some further call for comparative studies where census data are more reliable might be good.

We have added the following paragraph adopting this nice suggestion to the Discussion section (lines 257-260):

“The dominant root causes of rural population underrepresentation may vary significantly among countries, since the quality of the census and ancillary datasets is inhomogeneous. Further comparative studies in countries where census data are relatively robust would help improve our understanding of the strengths and limitations of the different population models in various contexts.”

[C3.6] Minor comments

1. line 27 - change degree to degrees
2. Supplementary material - section 1 - change scores to score

Thank you for noting these typos; we have fixed them.

Reviewer #4

The study finds significant discrepancies in global gridded population datasets when estimating rural populations living near a selection of global reservoirs. The datasets examined—WorldPop, GPW, GRUMP, LandScan, and GHS-POP—underestimate rural populations in these areas substantially, by 53%, 65%, 67%, 68%, and 84%, respectively.

[C4.1] *Will the work be of significance to the field and related fields? How does it compare to the established literature?*

The work is significant because it highlights a critical problem in widely-used population datasets. However it does overreach on its conclusions by applying lessons learned to rural areas as a whole rather than to the specific case of highly localized rural settlements consisting of small land areas. The study contributes to the literature by providing a global-scale validation of population datasets specifically in rural areas, which has been lacking.

Thank you for your comment. We are glad to hear you deem the study significant. We understand your concerns regarding the representativeness of the analysed 307 rural areas for rural areas as a whole, which was not sufficiently discussed in the original manuscript. To address your concerns, we have added a critical discussion on the limitations of the analysed sample. Moreover, we have looked further into several characteristics and properties of the sample (see our reply to comment C4.2).

But firstly, we would like to clarify what the analysed rural areas and their reported resettled populations actually represent: As described in the Methods section, and also briefly in the Introduction, the resettlement numbers for the analysed 307 areas **do not represent populations living near existing reservoirs**, which would be indeed a specific case. Instead, the data represents areas and **populations who lived in the areas before reservoirs were built** in these locations. Therefore, the reported populations did not have any relation to the reservoirs; they simply happened to be living in the exact areas that were later inundated due to reservoirs. This isolates the analysed sample from the specific case of reservoirs. For illustration, consider the example shown below in Figure R4: The analysed area is represented by the grey polygon, i.e. all pixels (and pixel portions if intersecting with the polygon boundary) located inside the grey polygon are taken into account in our analysis. The population data in Figure R4 shows people living inside the grey polygon, since there was no reservoir present yet at the reference year of the population map. The area was flooded only at a later point in time through the construction of a dam, resulting in the relocation of the people documented in our validation data. To improve clarity what exactly the validation data represent, we added Figure R4 to the Supplement and extended the descriptions in the Introduction and Methods sections, which read now as follows:

- Introduction, lines 52-60:
“As ground-truth data, we employ a combination of reported human resettlement numbers²⁷ and reservoir surface polygons²⁸ from 307 large dam construction projects spread over 35 countries (Figure 2). The resettlement numbers²⁷ were reported by national dam authorities and mostly stem from comprehensive on-the-ground impact assessments carried out during the planning and construction phase of the dam projects²⁹. The reservoir polygons²⁸, usually derived from satellite imagery, represent the areas inundated upon completion of the dams and thus provide the spatial extents from which the reported number of people²⁷ were displaced. For further details on the employed data and the methods used for comparing the ground-truth data to the gridded population datasets, please refer to Section 4.”
- Methods, lines 346-348:
“the numbers represent only physical resettlements from areas that were later inundated and occupied by reservoirs and do not include secondary displacements of people residing outside the reservoir areas, e.g. due to livelihood loss induced by the project”

Figure R4. Enlargement of Figure 1 in the manuscript, showing LandScan population data of the reference year 2000 around Na Hang Dam and Reservoir in Northern Vietnam, completed in 2008 and displacing 4000 people. Note the data resolution of 1 km for this dataset, and that only the populations inside the portions of the 1-km-cells falling inside the grey polygon are taken into account in the analysis (see Section 4.4 for details). This figure has been added as Supplementary Figure S1.

[C4.2] Does the work support the conclusions and claims, or is additional evidence needed? Is the methodology sound? Does the work meet the expected standards in your field?

The work does support conclusions concerning the more narrow and localized geographic scope of analyzing small settlements near a subset of global reservoirs, however the conclusions and claims of the authors that this causes a problem across rural areas as a whole are not fully substantiated. The scale of the analysis is of great importance here. In order to make sweeping statements and conclusions about the accuracy of these data in rural areas as a whole it would be necessary to analyze other rural contexts.

Thank you for your remarks; the scale and representativeness are indeed very important aspects to consider. In addition to improving clarity on the properties of the validation data and its specifics regarding reservoirs (see our reply to comment C4.1), we analysed four sample characteristics that could potentially affect representativeness (Table R1):

- Area size (see also Figure R2 and our reply to comment C3.2)
- Population number
- Population density
- Topography

We added the results and Table R1 to the Methods section, where we write:

“Among the 307 reservoirs, relatively small surface areas (1 – 25 km²) are most common, but also numerous larger areas up to about 4000 km² in size are included. Supplementary Figure S4 shows the size distribution of the reservoirs and illustrates that area size does not have an influence on the mean bias of the population datasets. In addition to area size distribution, we also analysed the distributions of population numbers, population densities, and altitudes of the 307 areas (Table 2). All four sample characteristics include a wide range of values, with standard deviations being larger than means and medians. This large spread in the data implies that our sample of 307 areas covers a great variety of contexts.”

Table R1. Statistical properties of the sample of 307 rural areas. Altitude data represents the ground elevation at the dam structure in metres above mean sea level, as reported by ICOLD. This table has been added to the manuscript as Table 2.

Characteristic of sample	Mean	Median	Std	Min	Max
Area [km ²]	73.3	9.0	278.3	1.0	3 645.6
Population [people]	11 818.2	1 864.0	56 362.7	0.0	900 000.0
Population density [people / km ²]	318.8	219.9	337.8	0.0	1 476.6
Altitude [mamsl]	481.3	270.0	609.2	7.0	4 250.0

However, despite the large variety among the 307 rural areas, we agree that the sample has important limitations: It is not representative in statistical terms, but a sample that was dictated by data availability. Still, our results provide a clear indication on rural populations being underrepresented in the analysed datasets, though further investigations are necessary to test whether the data is biased also in other rural areas that were not covered in our study. We describe these limitations now in the Discussion section (lines 310-315):

“Lastly, although the analysed set of 307 rural areas has a large variety (Section 4.2), it is not a representative sample in statistical terms that would allow us to evaluate population datasets for the whole global rural population. Nevertheless, our results provide a clear indication that rural populations tend to be underestimated by global population datasets. To further corroborate our findings for rural areas as a whole, we recommend additional validation studies using reference data from other rural contexts (e.g. resettlement data from surface mining activities).”

[C4.3] *Is there enough detail provided in the methods for the work to be reproduced?*

The study employs multiple datasets for cross-validation, and provides detailed explanations of the methods and adjustments used. There is sufficient detail in the methods section to reproduce the work.

We are very glad to hear, thank you.

[C4.4] *Remarks on code availability:*

Since this was a double blind assessment I did not follow the link to the github which would reveal the authors.

Thank you for the reminder. We created an anonymous code repository that can be accessed here: https://anonymous.4open.science/r/population_grid_assess/

References

Salmivaara, A. et al. Exploring the Modifiable Areal Unit Problem in Spatial Water Assessments: A Case of Water Shortage in Monsoon Asia. *Water* 7, 898–917 (2015)